# A New Approach of Soft Joint Based on a Cable-Driven Parallel Mechanism for Robotic Applications

**Luis Nagua** *,†, **Carlos Relaño** †, **Concepción A. Monje** and **Carlos Balaguer**

Robotics Lab of the Carlos III University of Madrid, Avda de la Universidad 30, Leganés, 28911 Madrid, Spain; crelgib@gmail.com (C.R.); cmonje@ing.uc3m.es (C.A.M.); balaguer@ing.uc3m.es (C.B.)
* Correspondence: lnagua@ing.uc3m.es; Tel.: +34-674859239
† These authors contributed equally to this work.

**Abstract:** A soft joint has been designed and modeled to perform as a robotic joint with 2 Degrees of Freedom (DOF) (inclination and orientation). The joint actuation is based on a Cable-Driven Parallel Mechanism (CDPM). To study its performance in more detail, a test platform has been developed using components that can be manufactured in a 3D printer using a flexible polymer. The mathematical model of the kinematics of the soft joint is developed, which includes a blocking mechanism and the morphology workspace. The model is validated using Finite Element Analysis (FEA) (CAD software). Experimental tests are performed to validate the inverse kinematic model and to show the potential use of the prototype in robotic platforms such as manipulators and humanoid robots.

**Keywords:** soft robotics; continuum mechanisms; modeling of complex systems; kinematic model of soft robots





## 1. Introduction

Soft robotics is a growing research area that has shown advantages over conventional robotics. In this area highly adaptive robots have been developed for soft interactions, providing greater security such as safe human-machine interaction. Compliance and adaptability of the soft structures are used for better efficiency and ability to interact with the environment [1]. Soft robotics is a new solution that covers the unmet need to perform tasks in unstructured and poorly defined environments, where conventional rigid robotics mainly seeks to be fast and accurate.

The advantages of soft robots allow for a wide variety of applications. However, this requires a paradigm shift in the methods of modeling, operation, control, materials and new designs to develop soft robots. The deformation property of soft robots is a restrictive element when using many of the most common conventional rigid sensors or other conventional control techniques [2].

Soft robotics is a subdomain of what is known as continuum robotics, it is defined by [3] as those robots with an elastic, continuously flexing structure and an infinite degree of freedom (DOF); and which are related to (but distinct from) hyperredundant robots, consisting of a finite number of many short, rigid links [4,5]. These models are usually more complex than traditional robot models, which have a small number of rigid links.

The incorporation of soft robotics into robotic systems comes mainly with two types of approaches [6]. One approach involves the use of compliant joints between different rigid links of the robot, while in another approach continuous soft robots are used, such as those mentioned above. This article explores this last type of design. Continuum soft robotic arms show features of soft robotics such as adaptability, high dexterity, and conformability to the external environment. However, they often cannot achieve the high rigidity and robustness required to handle objects or higher loads. Therefore, it is necessary to find a solution capable of providing the robustness of rigid arms and the versatility of soft

arms [7], which is one of the points addressed in this paper. From now own the term robustness will refer to the ability to cope with the action of external disturbances or loads that cause unwanted deformations in soft bodies, while providing sufficient stiffness.

Robots made of soft materials can generate complex behavior with simpler actuation by partially outsourcing control to their morphological properties and materials. That improves the active coupling of control, body and environment. Soft robots can be actuated in different ways, but the most common actuators are variable length tendons in the form of tension cables or shape memory alloys (SMA) [8,9], fluidic actuators such as pneumatic or hydraulic ones, and electro-active polymers (EAP) [2]. Other kinds of actuations focus on the morphological structure of the soft robot itself, as in [10], where the topological architecture of low-density soft robots is presented; Ref. [11], where a three-dimensional-printed soft origami rotary actuator is studied; or [12], which presents a soft origami tendon-driven actuator. For continuum robotic arms, pneumatic actuators are most used as they can continuously change their shape with a few DOF actuated [13]. However, arms with pneumatic actuators are usually less precise and difficult to control; less portable, since they rely on an external energy source such as a pneumatic compressor; and more expensive to maintain.

Many works have been carried out on the design and control of soft robots in recent years [14,15], but the state of the art shows that there are few approaches to soft robotic arms, either for integration into a manipulator robot or for use as a stand-alone manipulator. Some designs are based on soft silicone tentacles, as in [16] or [17]. Generally, this type of joint performs an instinctive gripping function that is actuated pneumatically or by cables, and its morphology does not allow its use in handling tasks that require greater precision and robustness. Nevertheless, other works such as [18–20] present soft robotic joint designs that combine a light weight and a high load-weight ratio. Others such as [21,22] present joints with an inflatable structure that can move through highly restricted environments by changing their three-dimensional structure.

Within the framework of soft servo-mechanical actuation, there are several examples, such as a cable-driven soft robot for cardiothoracic endoscopic surgery [23] or a practical 3D-printed soft robotic prosthetic hand [24]. In addition, servomechanically actuated soft limbs, which are closer to the proposal presented in this document, have been developed. An example is the neck developed by the DLR [25] and the soft robotic manipulator applying an adaptive algorithm [26] which includes a continuous silicone-based and tendon actuated mechanism. The RoboticsLab at the University Carlos III of Madrid has also developed a soft robotic neck [27,28] within the Humasoft project, with orientation and inclination capacity in the three-dimensional space and a large load capacity (with only 14 gr of weight, it can support up to 1 kg loads). Work has also been carried out on model identification of this robotic neck using different methods such as neural networks [29].

However, the use of those soft neck link designs cannot be generalized for soft robotic joints, as they do not meet the necessary robustness criteria. When working with different joint positions and orientations in 3D space, gravity comes into play depending on the orientation, and handling large loads can cause the joint to bend or break. Similarly, high stresses exerted on the tendons may seriously compromise the integrity of the actuation system.

These problems have motivated the approach presented in this paper, which is inspired by the soft robotic neck previously design by the authors [27,28]. Differently from that design, this new solution consists of a novel three-dimensional soft joint morphology based on asymmetric links. The joint is composed of a soft material that is flexible but robust. This material allows deformation to achieve bending movements, but prevents too complex deformations and undesired gravity effects. When the designed soft joint is bending, from a certain inclination angle and at certain orientation angles given by model measurements, a natural protection is provided by its own morphology, which limits maximum bending. Therefore, sufficient robustness is achieved to support different

loads throughout its positioning range in 3D space, while maintaining the advantages of its soft nature.

Furthermore, the proposed joint is scalable and adaptable to operational requirements in a modular and simple way. Therefore, joint properties, such as maximum bending angle or blocking bending, can be configured by modifying the morphological design and number of the links in the joint, or the distance between them, as well as increasing the number of DOF by concatenating joints.

Finally, this proposal is a low-cost construction, primarily designed by 3D printing and actuated by three motors that vary the length of tendons. Tendons are integrated within the morphology itself, which favors constant curvature and simplification of the model. Electromechanical action is proposed for the articulation, as opposed to other energy sources such as pneumatics or hydraulics. This feature allows the portability of the prototype and a greater integrability in any system (a robot, a humanoid, etc.), as well as more precise control and easier maintenance.

The rest of the paper is organized as follows: Section 2 introduces the soft joint design and prototype. It also shows its geometric design and includes the analysis of its characteristics and configurations. The section also shows the performance and assembly of the prototype and examines the properties of the material chosen for the joint morphology. Section 3 introduces the description of the mathematical model developed for the soft link, considering its workspace and the tendon length ratio that enables performance. The experimental tests carried out with the platform are described in Section 4, where the behavior of the soft joint is analyzed against different inputs and movements using two different tests. The discussion of the experimental results is presented in Section 5, and Section 6 concludes by highlighting the main achievements. This work is under a licensing process and the patent details are given in Section 7.

## 2. Design and Prototype of the Soft Joint

This section presents in detail the design and prototype of the soft joint.

### 2.1. Geometry

The soft joint has an asymmetrical morphology that allows its end tip to be positioned in the three-dimensional environment, robustly supporting high loads during its performance. Its design provides greater flexibility and a wider range of movement than a rigid joint. It consists of a series of links with asymmetrical prism morphology and circular section pitch. A triangular morphology is represented in Figure 1.

The small section and soft nature of the central axis of action, allow a greater bending capacity in all directions. The asymmetrical prismatic section provides the property of blocking and a natural protection, as well as the routing of the tendons for their action.

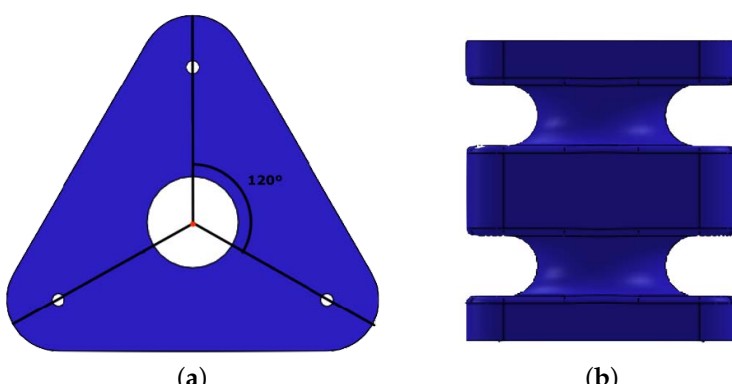

(a)          (b)

**Figure 1.** *Cont.*

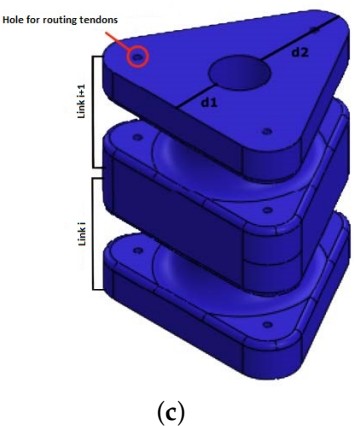

(**c**)

**Figure 1.** Triangular asymmetric geometry of the soft joint with two links in different views. (**a**) Top view showing the 120° angle relationship between the different tendon routing points. (**b**) Front view. (**c**) Perspective view, showing d1 and d2 distances defining asymmetry and holes for routing tendons.

The design performance is achieved by tendons that are routed through the asymmetric prismatic sections, as shown in Figure 2. It is possible to change the morphology of the prism and route the tendons through different points of these sections. This change would cause the variation of the forces and moments the joint is subjected to, therefore obtaining different kinematics and dynamics. By acting on the tendons, the joint can flex and orientate with two DOF.

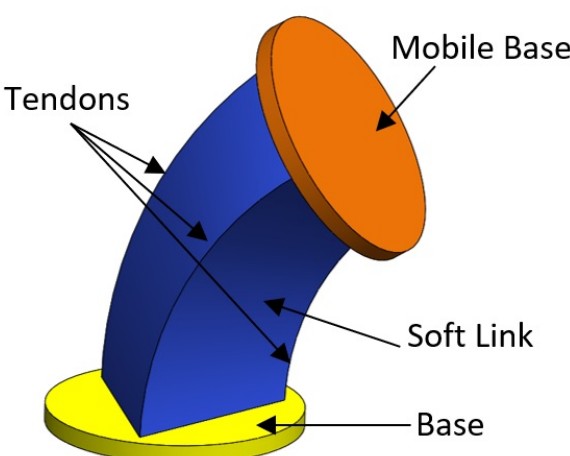

**Figure 2.** Conceptual design of the joint with its components: base, continuous soft axis, tendons for performance and tip (mobile base) of the soft joint.

One of the novel characteristics of this design is the natural morphological protection of the joint against large loads provided by the proposed asymmetrical morphology. An example of the triangular morphology are the two different configurations of extreme load:

- Configuration 1: Flexion towards one of the vertices of the triangle.
- Configuration 2: Flexion towards one of the edges of the triangle.

In configuration 1, protection when turning in the direction of one of the vertices is the most restrictive, as shown in Figure 3a. In the case of excessive bending, caused by high loads at the end of the joint or by control failures, the vertices contact each other. This produces a blocking curve of the structure that protects the joint from possible breakage due to wear or due to exceeding its elastic limit. This protection allows the joint to act with robustness and safety, especially in the regions of maximum flexion. In this configuration,

the action is achieved by a single tendon, which is routed through the vertices that form the bending curve.

Configuration 2 allows larger flexion of the joint, compared to Configuration 1, while also maintaining the natural protection of the joint. When the flexion is towards one of the edges of the triangle, the blocking curve has a smaller radius, as shown in Figure 3b. This is because the edges are closer to the central axis of rotation, as can be seen from the distance ratio $d1 < d2$ in Figure 1c. A larger bending occurs due to the fact that a larger bending angle is necessary before these edges contact each other and lock the joint structure. In this configuration, performance is achieved by the action of the two tendons that form the edge of the triangle where bending occurs.

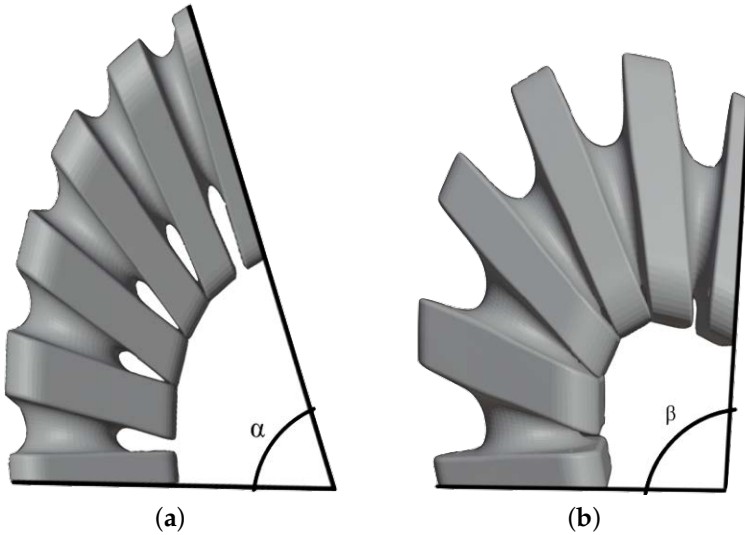

(**a**)  (**b**)

**Figure 3.** Different bending configurations. Relationship between bending angles: $\alpha < \beta$. (**a**) Flexion in configuration 1 has the lowest maximum bending angle. (**b**) Flexion in configuration 2 has the higher maximum bending angle.

*2.2. Actuation*

As mentioned above, there are several ways to operate soft robots. This paper focuses on operation by tendons of variable length using a winch coupled to a motor shaft. Tendon lengths must be translated into motor angular positions. $L_o = 0.2$ m is the length of the tendons when the joint is at rest position, and $L_i$ is the target tendon length. The linear displacement is transformed into an angular displacement by the length of the arc formed by the circumference of the winch for a certain angle (Figure 4), following the equation below:

$$\Omega = \frac{(L_o - L_i)}{R} \tag{1}$$

$R$ is the radius of the winch where the tendon is wound or unwound, in this case 9.3 mm, and $\Omega$ is the angle that provides that displacement.

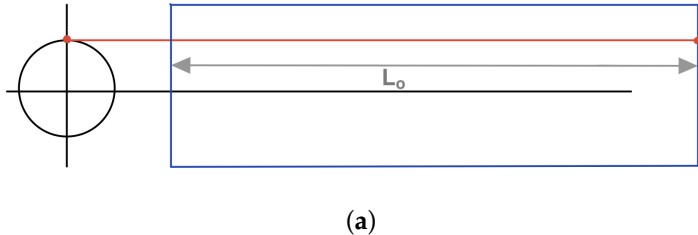

(**a**)

**Figure 4.** *Cont.*

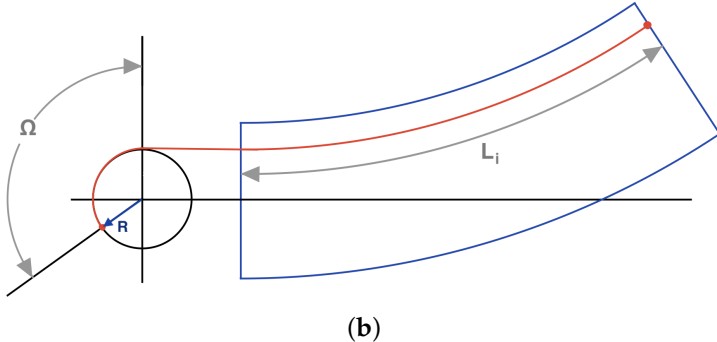

**(b)**

**Figure 4.** Diagram depicting winch winding based on radius and angle. $L_o - L_i$ is the distance for tendon winding, and $\Omega$ and $R$ are the angle and the radius, respectively. (**a**) Tendon and soft joint prior winch actuation. (**b**) Tendon and soft joint when the winch is operating, with the radius R, and the angle $\Omega$.

### 2.3. Prototype

To choose the soft joint operation, a test platform was designed. The goal is that the rest position of the joint is horizontal. Three motors will be used to operate the joint by tendons, each of which will wind the three tendons (Figure 5).

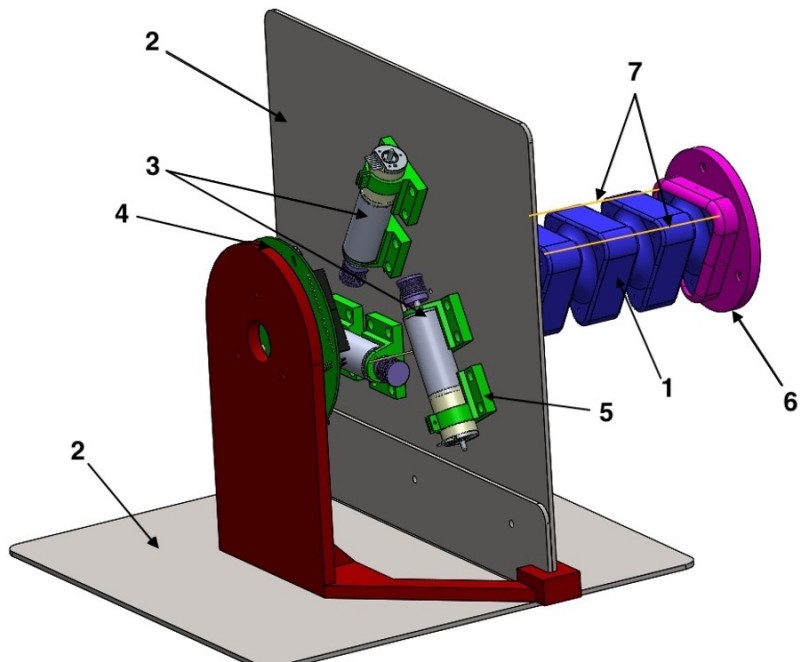

**Figure 5.** The elements of the platform are the soft joint (**1**), a metal base (**2**), motors (**3**), electronic elements to feed and control the motors (**4**) and other connective elements such as motor supports (**5**), joint bases (**6**) and tendons (**7**).

The fixing base is made up of two 3 mm thick metal plates, to be strong enough to support the test loads. The motors used for the drive are Maxon EC-max 22. The motors are controlled by Technosoft's Intelligent Drives iPOS 4808 MX, which communicate with the PC via busCAN.

Connecting elements have been printed on a 3D printer Creatbot600 pro and Zmorph from PLA (Polylactic acid) material. They are two bases for fastening the soft joint with the metal base, a platform for fastening the electronic elements, three motor fasteners with the metal platform and three winches that are attached to the motor shaft and the tendons,

made of polyester thread, for the activation of the joint. The designed soft joint has been built by 3D printing from NinjaFlex using a Creatbot600 pro printer (Figure 6).

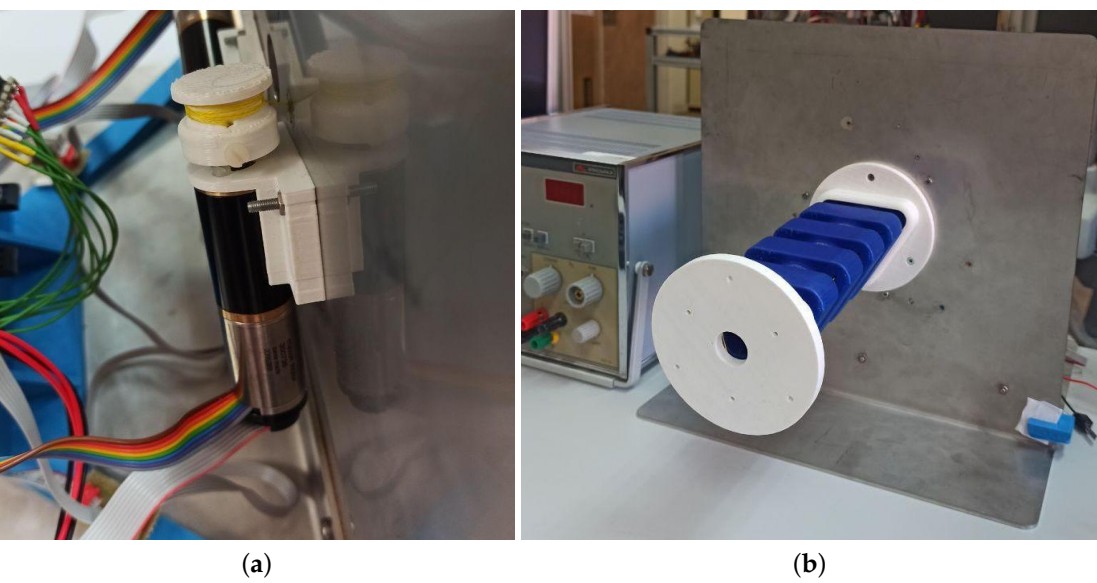

|     |     |
|:---:|:---:|
| (**a**) | (**b**) |

**Figure 6.** (**a**) Motor connected to the winch to wind the tendons for joint actuation. (**b**) Soft joint on the test platform.

### 2.4. Material Properties and Tests

One of the most important features when prototyping a soft robotic joint is the choice of material. This design uses NinjaFlex® 3D Printing Filament, a flexible polyurethane material for Fused Deposition Modeling (FDM) printers. This 3D printing manufacturing method and this material were chosen for their ease to use and for allowing variations in percentage or filling patterns of the soft joint body.

The mechanical properties of this material make it a good choice for the purpose of the prototype (Table 1). Its flexibility allows the joint to bend but, at the same time, it is rigid enough to prevent big deformations and resist loads.

**Table 1.** Mechanical properties of NinjaFlex®.

| Mechanical Properties | Value |
|:---:|:---:|
| Young's modulus | 12 MPa |
| Hardness | 85 Shore A |
| Poisson Ratio | 0.48 |
| Density | 1040 Kg/m$^3$ |

The soft joint design was analyzed in SolidWorks software, which applies a non-linear finite element study on the material. The prototype was modeled as a simple cantilever beam (one of its ends is fixed and a force is applied to its free end). This allows an efficient testing of the design under stresses and strains.

To simplify the simulation, the joint was assumed to be a completely filled solid except for the inner channel, and to simulate the assembly of the real prototype, the soft joint model was assembled including its two support pieces, one at each end.

After the design phase, the prototype was 3D printed using NinjaFlex material with 30% infill. The experiments were performed with this specific prototype.

The model in SolidWorks was tested under different conditions. First, a no-load test was performed on the soft joint, by only simulating gravity and fixing one of the ends, as shown in Figure 7, with the red arrow representing the orientation of the gravity action in the simulation.

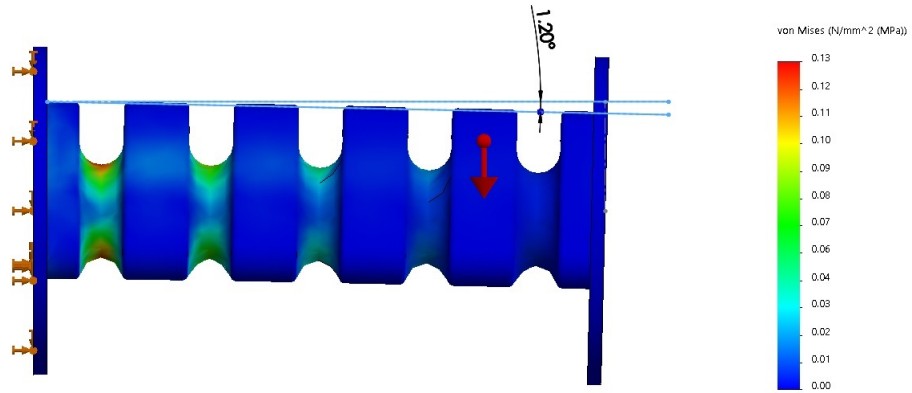

**Figure 7.** No-load simulation of soft joint.

One intended use of this soft joint is as a manipulator able to support different loads. Therefore, a second simulation was carried out with a rectangular prism with a fixed mass of 500 gr, homogeneously distributed. This prism represents the weight of the robot gripper in the simulation, Figure 8. In addition, a 10 Newtons downward force is applied to the end effector, simulating an external weight of 1 kg and causing a higher end torque. The simulation shows a deflection of 7.38° and a maximum deformation of 0.75 MPa.

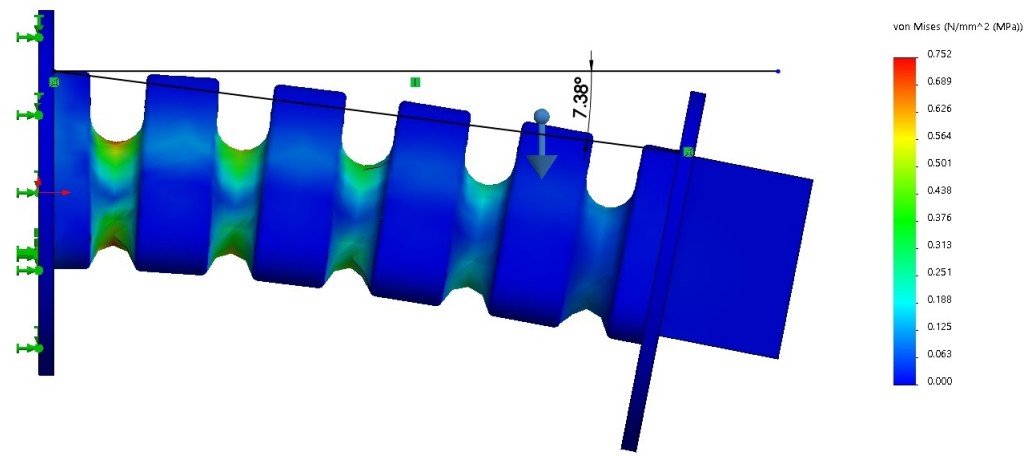

**Figure 8.** Simulation of soft joint with a 500 g prism and a 10 N downward force at the free end.

Additionally, another stress study was carried out to check if the yield strength of Ninjaflex is not exceeded. It was noted that when applying 60 N force at the end of the soft joint, as shown in Figure 9, a bending angle of 60° was reached and the maximum deformation was 2.9 MPa. Therefore, a no permanent deformation is confirmed when the soft link reaches an inclination angle of 60°.

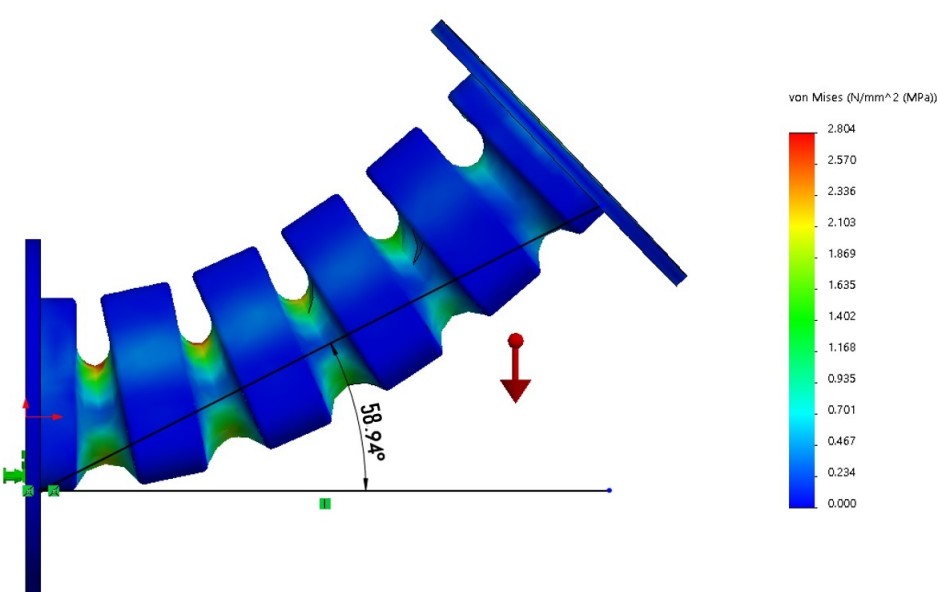

**Figure 9.** Simulation of soft joint with a 60 N upward force applied on the free end.

## 3. Mathematical Model of the Soft Link

The position of the soft joint is defined as the combination of orientation and inclination, where inclination is the curvature angle of the joint, and orientation is the angle of the plane perpendicular to the base that contains that curvature. It achieves two DOF of flexion from the three tendons, thus the position depends on the distance of the tendons and their combination. Therefore, a mathematical model of the joint has been created to obtain the theoretical distances of the tendons required for a specific position of the end of the joint. This angle is assumed to be zero when it coincides with the Y axis, and the actuators are named counterclockwise as this angle increases, Figure 10a.

### 3.1. Calculation of Tendon Lengths

The robot inputs are one inclination value, $\theta$, and one orientation value, $\psi$, and the outputs will be tendons lengths:

$$L_i = [L_1 \; L_2 \; L_3]' \tag{2}$$

Inverse kinematics was used to calculate tendon lengths for the target end position. It is important to point out that unlike works such as [27] or [25], this design does not have the tendons in the open air, but the performance of the tendons is embedded within the morphology of the soft joint itself. This makes the length of the tendons not straight, but rather the tendons project the curvature of the soft joint, thus having a curvature similar to that of the joint. Therefore, $L_i$, the lengths of the tendons form an arc between both ends of the joint, Figure 10b.

Thus, tendons and joint are considered robots shaped by continuously bending actuators, such as those described by [30,31], where a pneumatic actuation is usually used, considering joint curvature and tendon curvature as a continuous curvature. The equations shown in [3] are adapted to this specific morphology case.

An angular-curved approach is used, with the inclination and orientation parameters. The lengths of the tendons $L_i$ depend on both inclination and orientation angles. The length of the joint, $L$, remains constant in its central fiber at all times, regardless of the curvature; and the distance, $a$, of the tendons from the center of the joint section, remains constant, too (Figure 10b). For this morphology, $a$ measures 0.035 m, $L$ measures 0.2 m. The actuator for tendon 1 is placed at $v_1 = \frac{\pi}{2}$ radians, tendon 2 is placed at $v_2 = \frac{7 \cdot \pi}{6}$ radians and tendon 3 is placed at $v_3 = \frac{10 \cdot \pi}{6}$ radians.

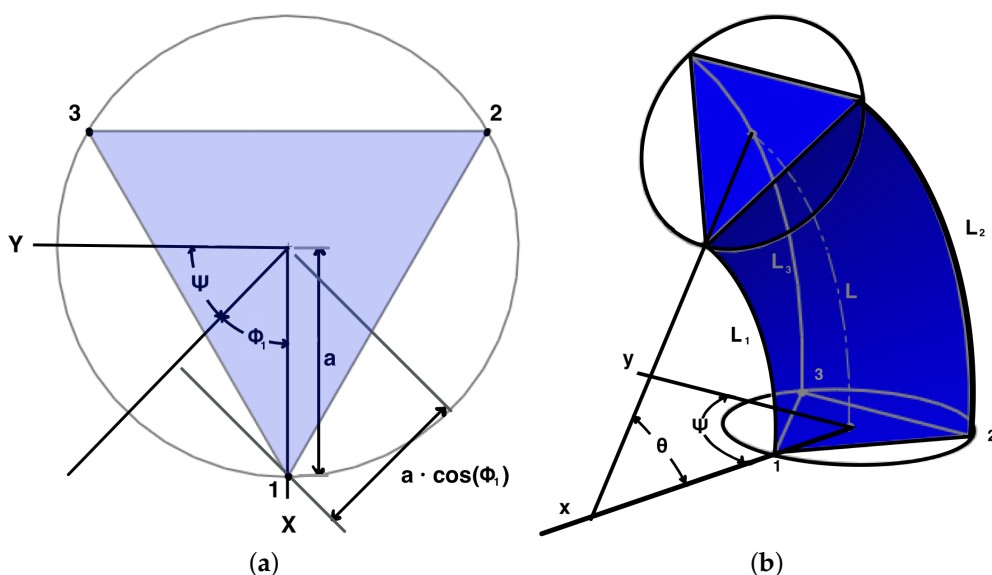

**Figure 10.** (**a**) Base projection of the soft joint, $\psi = 45°$, for the representation of orientation, distances and numbering of tendons. (**b**) Three-dimensional representation of the joint with $\theta = 45°$ orientation and $\psi = 90°$ inclination. Note the different curvatures for the soft joint and for each tendon $L_i$.

As previously discussed, it can be determined that $L$, the central fiber length of the soft joint, is constant independently of the inclination angle. Tendon lengths are calculated through the arc equations, due to the assumption of constant curvature. The radius $r$ of the curvature $L$ is determined as $L = r \cdot \theta$, where $\theta$ has a value in radians. As the central fiber and tendons move, they move in the direction given by the angle of orientation, and by projecting the arcs and radii, the representation in Figure 11 is obtained. Therefore, $L_i$ can be determined as $L_i = r_i \cdot \theta$, where $r_i = r - a \cdot cos(v_i - \psi)$, resulting in the following equations:

$$L_1 = L - \theta \cdot a \cdot cos(v_1 - \psi) \tag{3}$$

$$L_2 = L - \theta \cdot a \cdot cos(v_2 - \psi) \tag{4}$$

$$L_3 = L - \theta \cdot a \cdot cos(v_3 - \psi) \tag{5}$$

Hence, $\phi_i$ is the angle between orientation, which is the plane containing the curvature, and the plane of tendon location, $i$. This angle $\phi_i$ depends on the configuration of the orientation and the number of actuators. The relationship of each tendon with the orientation is as follows:

$$\phi_1 = v_1 - \psi \tag{6}$$

$$\phi_2 = v_2 - \psi \tag{7}$$

$$\phi_3 = v_3 - \psi \tag{8}$$

A generic equation is obtained for lengths:

$$L_i = L - \theta \cdot a \cdot cos(\phi_i) \tag{9}$$

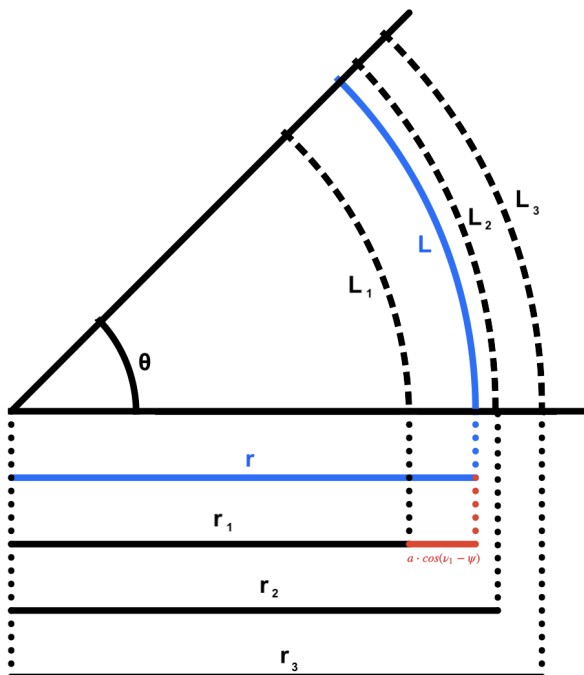

**Figure 11.** Representation in the perpendicular view of the orientation plane formed by the orientation angle $\psi$ and an inclination angle $\theta$. It can be seen that the projection of the radii of the constant curvature of the soft joint. The central fiber curvature $L$ and its corresponding radius $r$ are represented in blue. The arcs of tendons $L_i$ are represented by dashed black lines, and their corresponding radii $r_i$ by continuous black lines. The difference between $r$ and $r_i$ is represented by a red line whose distance for each tendon is given by equation $a \cdot cos(\nu_i - \psi)$.

### 3.2. Calculation of the Blocking Angle

The proposed morphology is designed with a blocking mechanism that protects or strengthens it at certain angles of inclination and orientation, and that must be parameterized in the kinematics. The angle of inclination at which the blocking occurs depends on the space between the triangular sections, where $H_s$ is the height of the point of contact with the bending center of the link, and $D_s$ is the distance from the point of contact with the bending center of the link, as shown in Figure 12. However, this distance $D_s$ is not a constant parameter as it would be if the sections were circular. The blocking angle depends, in this asymmetric triangular design, on the distance $D_s$, which varies according to the orientation being a maximum value when the point of contact is the vertices of the triangle and a minimum value when the point of contact is the center of the edges of the triangle.

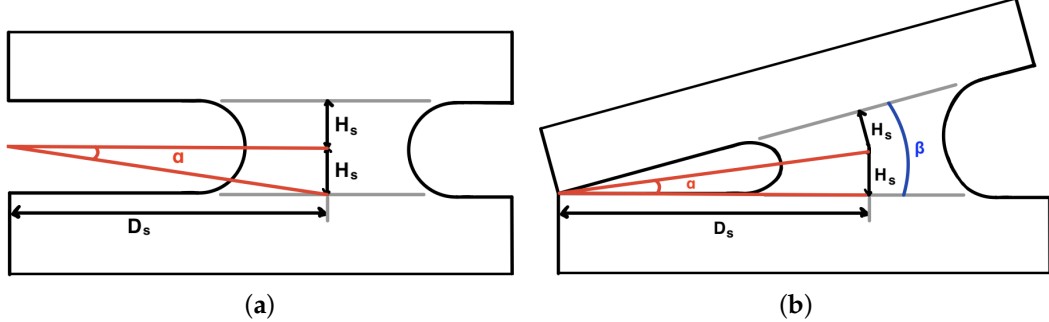

| (a) | (b) |

**Figure 12.** (**a**) Diagram showing the link bending with the joint at rest. (**b**) Bending of the beta link at the point where the morphology makes the blocking contact.

From the values $H_s$ and $D_s$ the angle $\alpha$ is obtained as:

$$\alpha = \arctan\left(\frac{H_s}{D_s}\right) \tag{10}$$

This angle is formed as the bisector of the blocking angle. The blocking angle of a link, $\beta$, is given as the double of alpha and it is obtained from the following equation:

$$\beta = 2 \cdot \alpha = 2 \cdot \arctan\left(\frac{H_s}{D_s}\right) \tag{11}$$

$H_s$ has a fixed value (in our case, 8 mm) while $D_s$ varies according to the orientation. To calculate $D_s$, we estimated the maximum, $\epsilon_{max}$, and minimum, $\epsilon_{min}$, possible distances with this morphology (40 mm and 25 mm, respectively), and the angles between them, $\psi_{dif} = 60°$. Knowing the orientation angles where the maximum and minimum occur, it can be parameterized according to a factor such that:

$$\frac{\epsilon_{max} - \epsilon_{min}}{\psi_{dif}} = 0.25 \tag{12}$$

Based on this factor, we know how the distance between the minimum and the maximum varies for each degree for $D_s$.

Once the theoretical blocking angle, $\beta$, is estimated for each link according to the orientation, we can calculate the final joint angle, $\Gamma$, when blocking occurs.

The final angle depends on the number of links within the joint, $N$, such that:

$$\Gamma = \beta \cdot N \tag{13}$$

### 3.3. Representation of the Workspace

Joint kinematics will block angles greater than the total blocking joint angle, creating an asymmetric workspace. X, Y and Z axes represent the soft joint final position in meters. The soft joint fixed base is at position $[0, 0, 0]$. Maximum Z value is 0.2 m when the joint is at rest. As the soft joint flexes, Z value decreases. X and Y values are the projection of the joint end position on the base plane. They are zero at resting position, and change with flexion. Therefore, the designed soft joint does not perform the same bending angle, both being performed in the same plane.

If this is done for different planes, we obtain a 3D mesh of $'*'$ marks. The surface of a non-complete sphere is obtained, as seen in Figures 13 and 14. This allows knowledge of where the end will be and how the soft joint will move with respect to the fixed base.

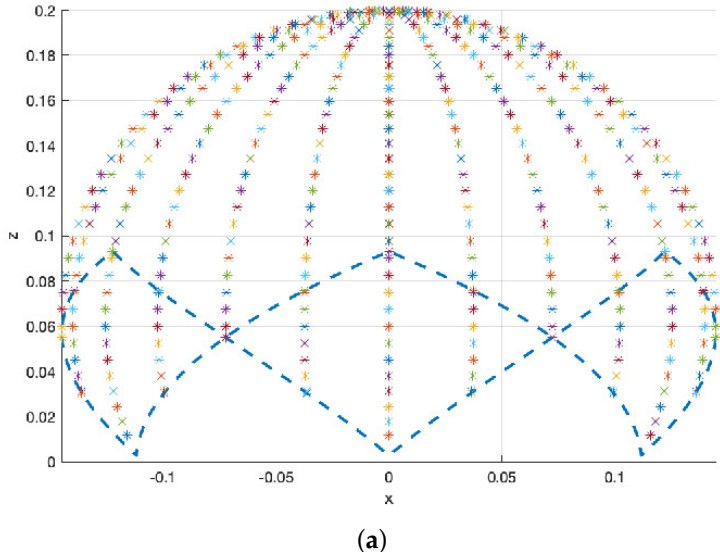

(a)

**Figure 13.** *Cont.*

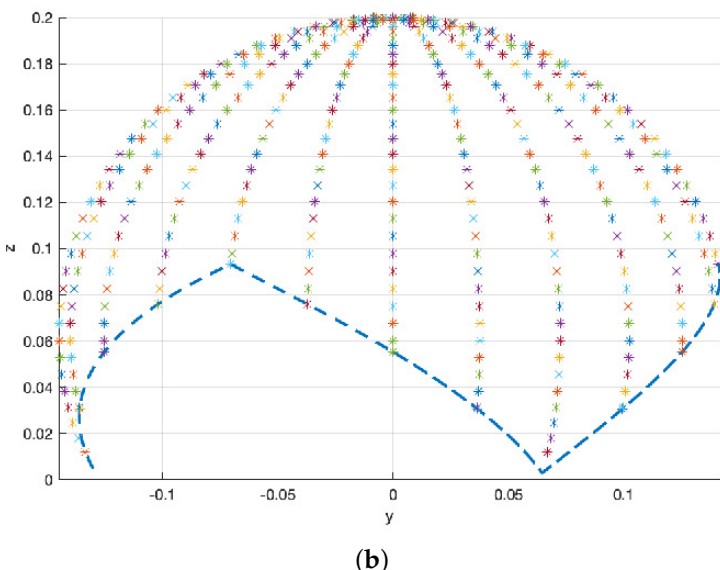

**(b)**

**Figure 13.** Diagram of soft joint end positions, represented by '∗' marks, at different orientation planes, every 15 degrees, and for every 5 inclination degrees. The dashed line indicates the flexion limits of the soft joint for each orientation. The soft joint cannot reach further positions due to blockages. (**a**) Front view. (**b**) Side view.

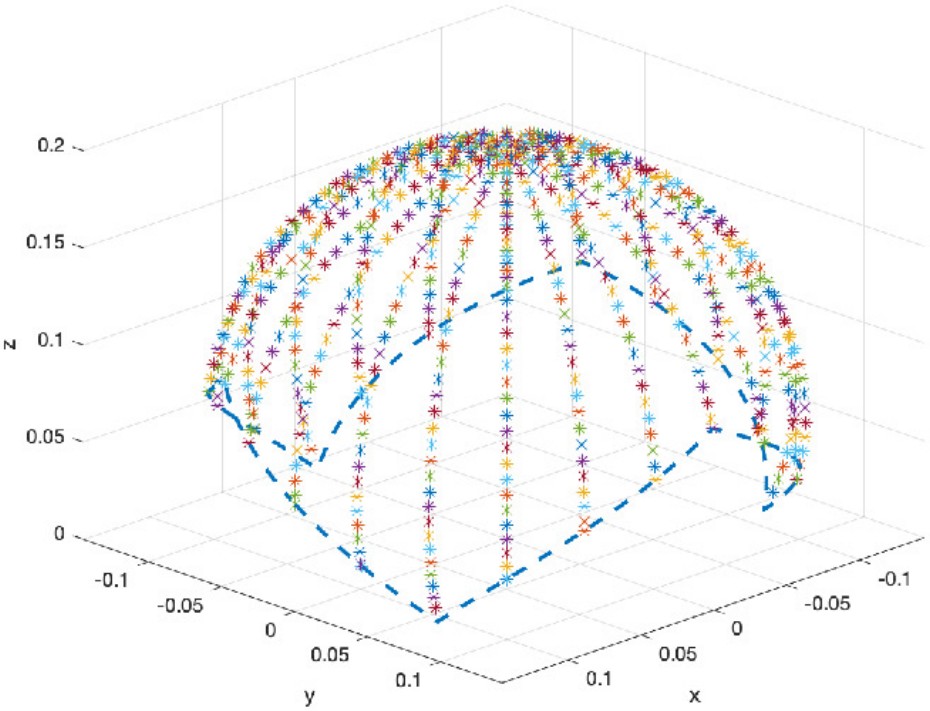

**Figure 14.** Perspective view of the soft link end positions represented by '∗' marks, at different orientation planes, every 15 degrees, and for every 5 inclination degrees. The dashed line indicates the flexion limits of the soft joint for each orientation. The soft joint cannot reach further positions due to blockages.

### 3.4. Representation of Variations in Tendon Lengths

Once tendon distances are adjusted to the joint kinematics, with the blocking angle restrictions, distance changes for each tendon can be represented as inclination and orientation input angles vary. Figure 15 shows tendon lengths according to inclination and

orientation variations, the restrictions imposed by the design morphology, 0 to 359° orientation degrees and 0 to 170° inclination degrees, and the final length in meters.

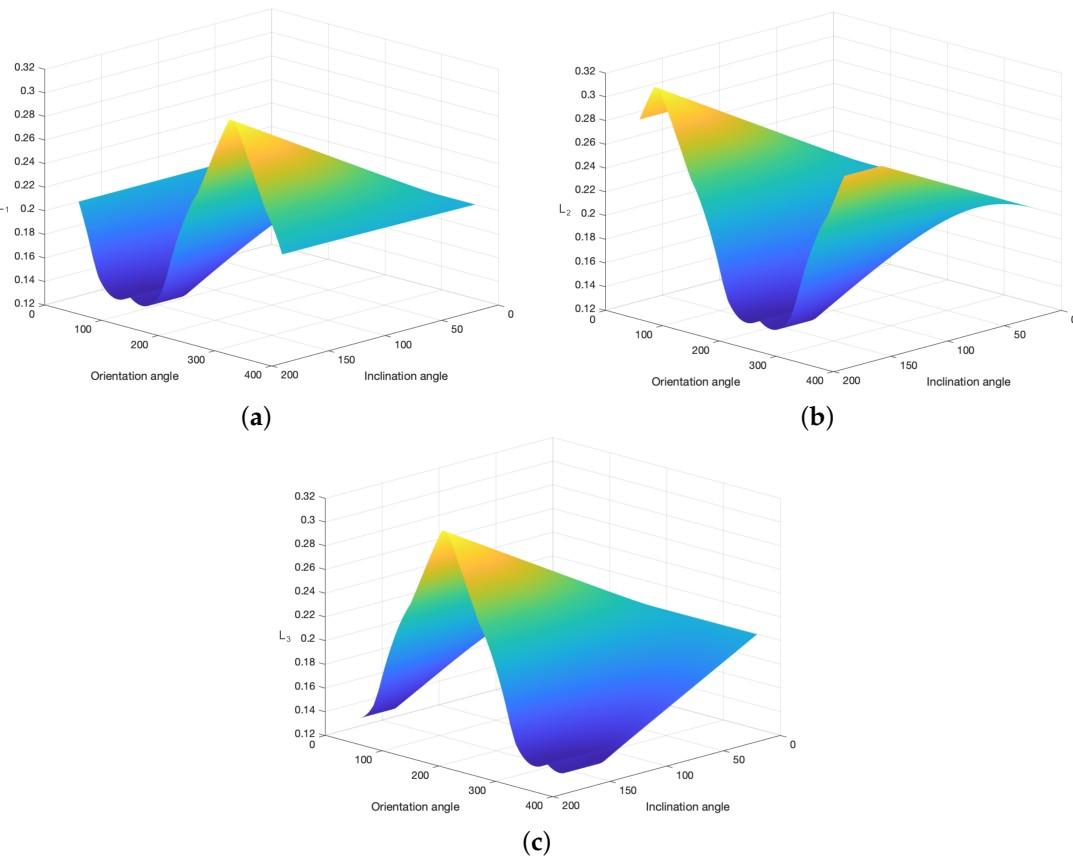

**Figure 15.** Representation of each tendon length variation for all possible values of inclination and orientation of the soft joint. The initial length of each tendon is 0.2 m (rest position). A color range from yellow to dark blue is used to show the variation from the highest tendon length value (yellow) to the lowest value (dark blue). (**a**) Length $L_1$ corresponding to tendon 1, (**b**) Length $L_2$ corresponding to tendon 2, (**c**) Length $L_3$ corresponding to tendon 3.

These graphs show how each tendon $L_i$ varies according to inclination and orientation. The higher the inclination, the higher the variation of tendon length with changes of orientation. For a fixed inclination, when the orientation changes, as in a rotational movement, the tendon length increases and decreases in a sinusoidal shape, with the orientation corresponding to a maximum, a minimum or the initial length value. Due to the soft joint blockages, from certain degrees of inclination, the variation of tendon lengths is not sinusoidal, and, for certain orientation angle ranges, the length remains fixed.

*3.5. Direct Kinematics*

A direct kinematics is also provided through the works collected in [3]. This kinematics allows us to know the inclination and orientation for the input values $L_1$, $L_2$ and $L_3$. These equations assume that the curvature is constant throughout the flexible body.

$$\psi = \arctan\left(\frac{\sqrt{3}(l_2 + l_3 - 2l_1)}{3(l_2 - l_3)}\right) \tag{14}$$

$$\theta = \frac{2\sqrt{l_1^2 + l_2^2 + l_3^2 - l_2l_1 - l_2l_3 - l_1l_3}}{a(l_1 + l_2 + l_3)} \tag{15}$$

### 3.6. Simulation of the Model

Using the above equations, the mathematical model can be represented by simulation. From the inputs, inclination and orientation, the inverse kinematics is made, and the linear displacement of the tendons is calculated. Those values are turned into and angular displacement for each motor. The motor encoders can be used as sensors to measure the real angular motor position and close the control loop.

The motor models are represented as a function using the values from the motor datasheet. Following a general control diagram, where $K$ is the motor speed constant in rpm/V, and $\tau$ is the mechanical time constant in seconds, we obtain the transfer function G(s) [32], such that:

$$G(s) = \frac{K}{s(1 + \tau s)} = \frac{352}{s(1 + 0.00875s)} \tag{16}$$

For the simulation, a control loop is created in Simulink Matlab, in which the input values are entered interactively, Figure 16. The tendon lengths for these inputs are obtained through a Matlab function that has been designed from Equation (9), called "Inverse Kinematics", Algorithm 1.

---

**Algorithm 1** Inverse kinematics.

1:    *input:* $\theta, \psi$
2:    *internal constant:* N, L, a
3: **procedure**
4:     $\beta \leftarrow$ *block-angle-equation*$(\psi)$
5:     **if** $\beta < \theta/N$ **then**
6:        $\theta = \beta \cdot N$
7:     $\phi_i \leftarrow$ *phi-equation*$(\psi)$
8:     $L_i \leftarrow$ *length-equation*$(L, a, \phi_i)$
9:     **return** $L_i$

---

The three values of $L_i$ returned by the inverse kinematics block are used to obtain the target $\Omega$ (target angular position of the motors), using the "$L_i$ to Omega" function block described by Equation (1), Algorithm 2.

---

**Algorithm 2** $L_i$ to Omega.

1:    *input:* $L_i$
2:    *internal constant:* L, r
3: **procedure**
4:     $\Omega \leftarrow$ *Omega-equation*$(L_i, L, r)$
5:     **return** $\Omega$

---

From these target $\Omega$ values, the motor control loops return the current $\Omega$ values. The direct kinematics is performed using the "Direct Kinematics and 3D representation" function block defined by Equations (14) and (15), Algorithm 3. The current inclination and orientation of the free end through the simulation are obtained. This function block also provides the position of the simulated soft joint represented in a 3D space, Figure 17.

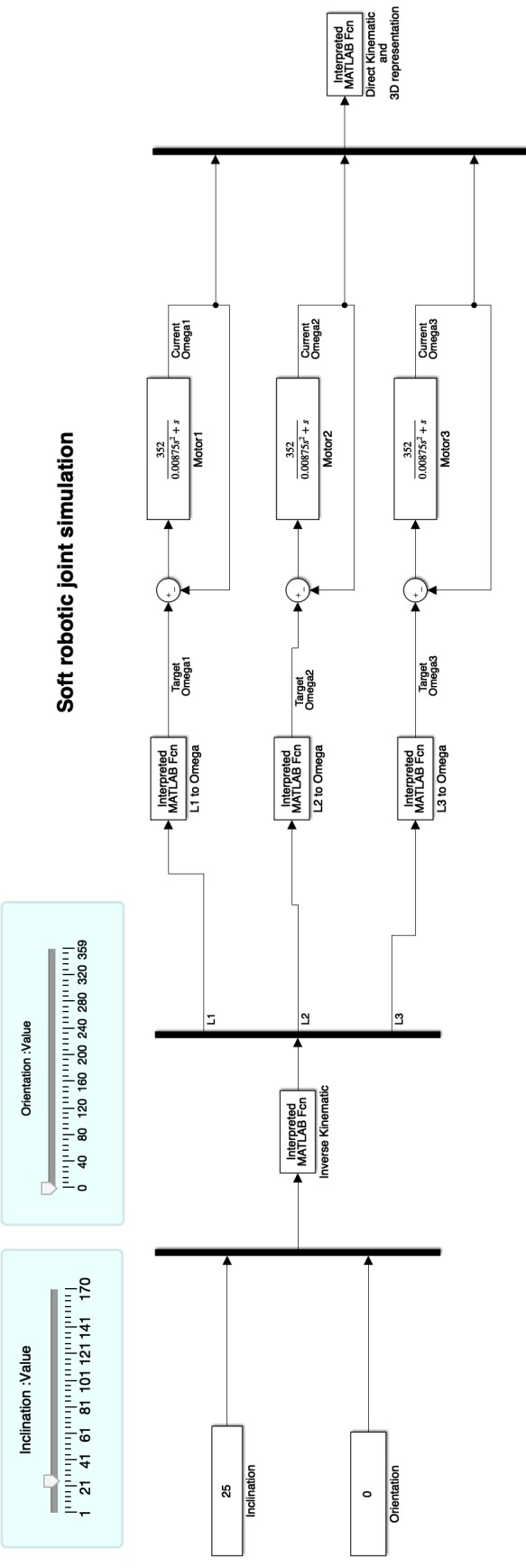

**Figure 16.** Simulink schematic for the soft joint model simulation.

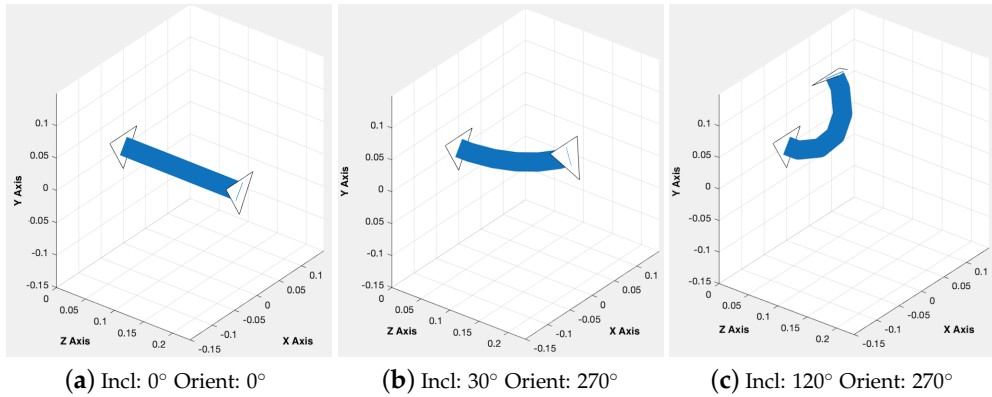

(**a**) Incl: 0° Orient: 0°   (**b**) Incl: 30° Orient: 270°   (**c**) Incl: 120° Orient: 270°

**Figure 17.** Three-dimensional representation of the simulation of the soft joint for different input values.

---

**Algorithm 3** Direct Kinematics and 3D representation.

---

1:   *input:* $\phi_i$
2:   *internal constant:* L, r, a
3:   **procedure**
4:      $L_i \leftarrow$ *Omega-equation-inverse*$(\phi_i, L, r)$
5:      $\theta_{Simu} \leftarrow$ *Direct-Kinematics-$\theta(L_i, a)$*
6:      $\psi_{Simu} \leftarrow$ *Direct-Kinematics-$\psi(L_i)$*
7:      *Draw-simulation*$(\theta_{Simu}, \psi_{Simu})$
8:      **return** $\theta_{Simu}, \psi_{Simu}$

---

## 4. Experimental Tests

The soft joint assessment is performed through two types of experimental tests. These tests allow us to evaluate motion performance and kinematics model accuracy, based on the error between the target end position and the real end position of the soft joint. A video showing these tests performance can be viewed at https://vimeo.com/537605947 (accessed on 10 May 2021).

Data were collected from the tests in two ways. Position data from motor encoders provided information on inclination and orientation through the direct kinematics. Data from the inertial sensor 3DM-GX5-10 IMU, the yaw, roll and pitch data, were transformed into inclination and orientation data for comparison with references.

### 4.1. Test 1

Test 1 consists of a bending movement towards a fixed inclination angle, in each of the four orientations: 0°, 90°, 180° and 270°. This test shows how the joint starts in a resting position, performs the action and then returns to the resting position before it bends at the next orientation. The resting position is 0 degrees of inclination and orientation. Tests were performed for 30°, 45° and 60° inclination and results are shown in Figure 18 for the encoder data and Figure 19 for the sensor data.

Test 1—Encoder data

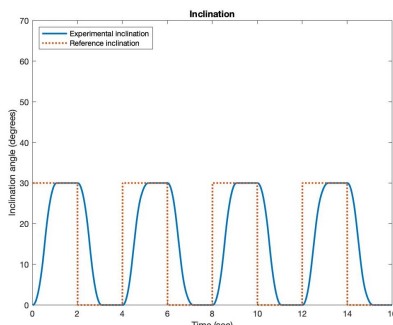

(**a**) Inclination versus time through encoder measurements for 30° inclination.

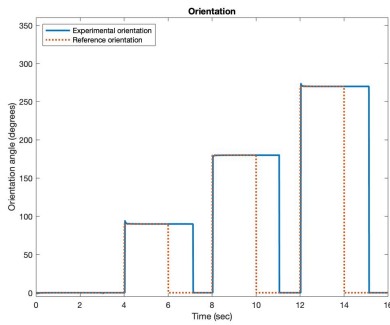

(**b**) Orientation versus time through encoder measurements for 30° inclination.

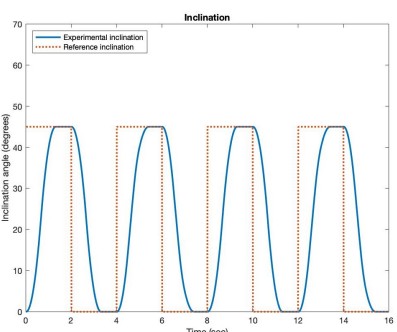

(**c**) Inclination versus time through encoder measurements for 45° inclination.

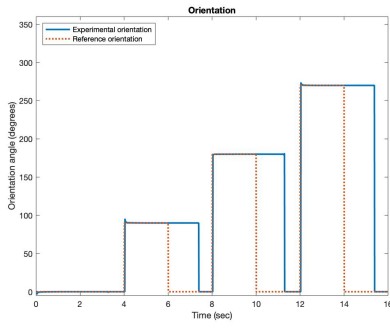

(**d**) Orientation versus time through encoder measurements for 45° inclination.

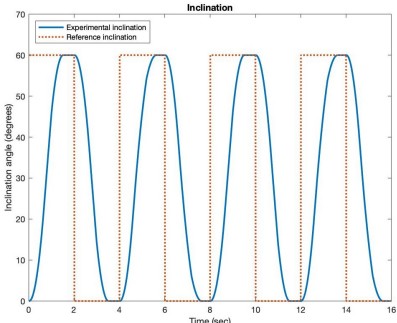

(**e**) Inclination versus time through encoder measurements for 60° inclination.

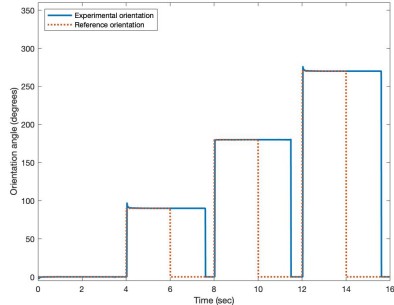

(**f**) Orientation versus time through encoder measurements for 60° inclination.

**Figure 18.** Test 1—Encoder data. Fixed 30°, 45° and 60° inclination for four orientations: 0°, 90°, 180° and 270°. The blue line is the experimental data obtained from the encoder and the orange dotted line is the reference.

Test 1—Sensor data

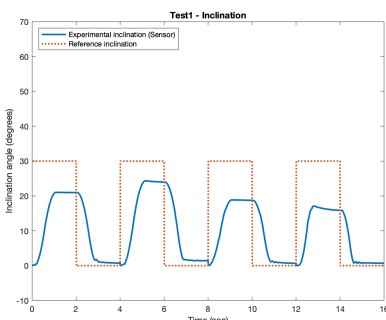

(**a**) Inclination versus time through inertial sensor measurements for 30° inclination.

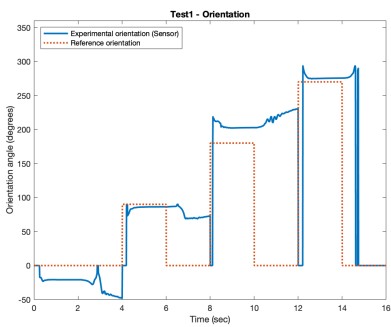

(**b**) Orientation versus time through inertial sensor measurements for 30° inclination.

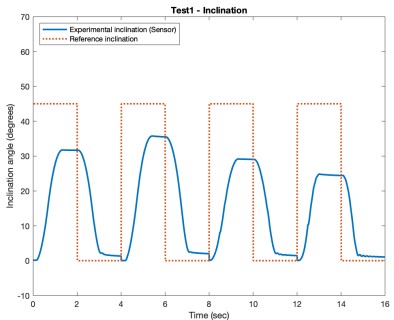

(**c**) Inclination versus time through inertial sensor measurements for 45° inclination.

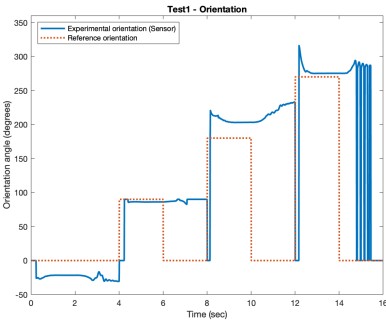

(**d**) Orientation versus time through inertial sensor measurements for 45° inclination.

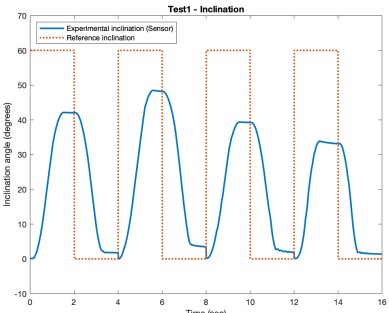

(**e**) Inclination versus time through inertial sensor measurements for 60° inclination.

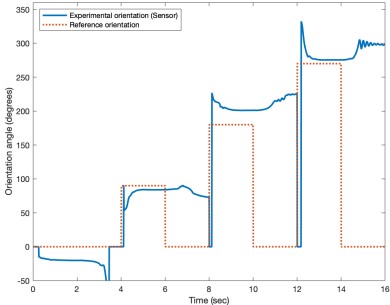

(**f**) Orientation versus time through inertial sensor measurements for 60° inclination.

**Figure 19.** Test 1—Sensor data. Fixed 30°, 45° and 60° inclination for four orientations: 0°, 90°, 180° and 270°. The blue line is the experimental data obtained from the encoder and the orange dotted line is the reference.

*4.2. Test 2*

Test 2 consists of a 360° rotation for a given inclination. This rotation starts in a resting position and is performed by increasing the orientation value by one degree every 0.1 s, starting from 0°. When the rotation is complete, it returns to the resting position. The test was performed for 30°, 45° and 60° inclination and results are shown in Figure 20 for the encoder data and Figure 21 for the sensor data.

Test 2—Encoder data

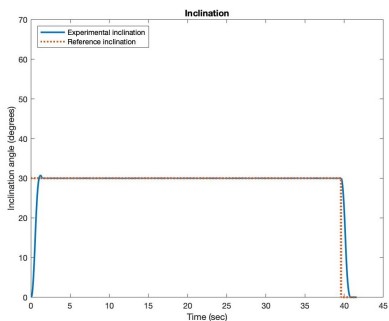

**(a)** Inclination versus time through encoder measurements for 30° inclination.

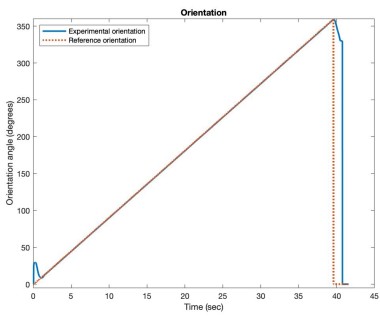

**(b)** Orientation versus time through encoder measurements for 30° inclination.

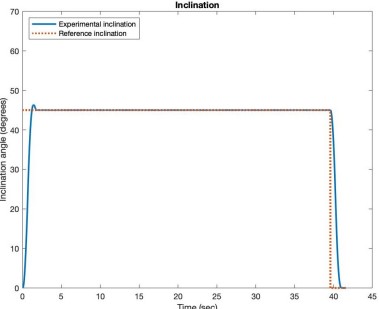

**(c)** Inclination versus time through encoder measurements for 45° inclination.

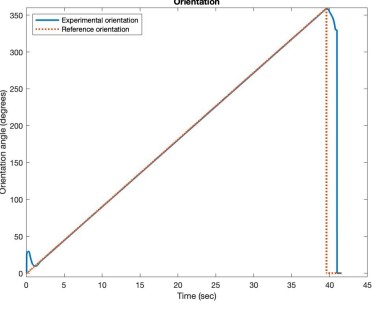

**(d)** Orientation versus time through encoder measurements for 45° inclination.

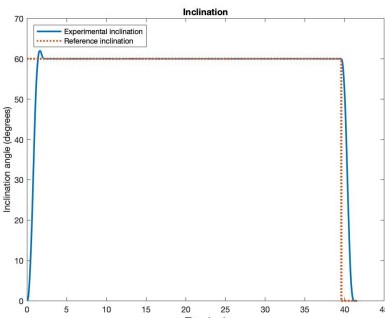

**(e)** Inclination versus time through encoder measurements for 60° inclination.

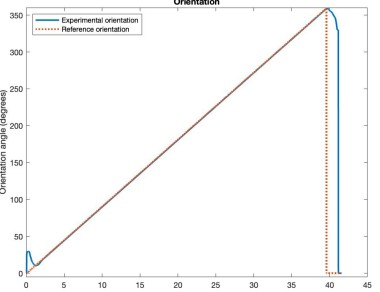

**(f)** Orientation versus time through encoder measurements for 60° inclination.

**Figure 20.** Test 2—Encoder data. Fixed 30°, 45° and 60° inclination for a 360° rotation. The blue line is the experimental data obtained from the encoder and the orange dotted line is the reference.

Test 2—Sensor data

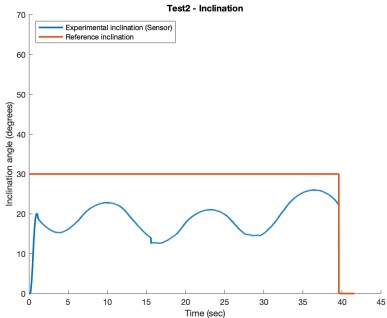

(**a**) Inclination versus time through inertial sensor measurements for 30° inclination.

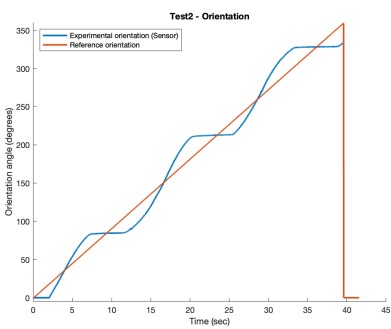

(**b**) Orientation versus time through inertial sensor measurements for 30° inclination.

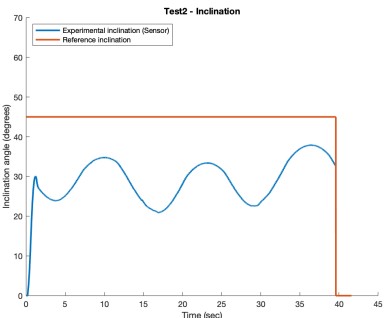

(**c**) Inclination versus time through inertial sensor measurements for 45° inclination.

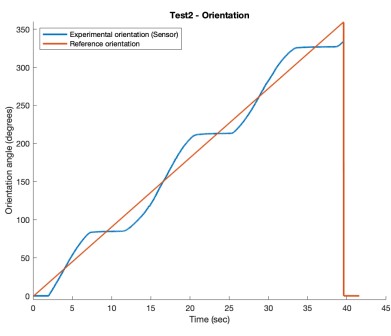

(**d**) Orientation versus time through inertial sensor measurements for 45° inclination.

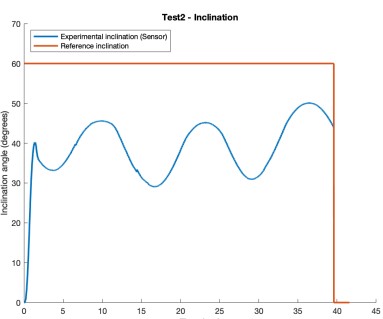

(**e**) Inclination versus time through inertial sensor measurements for 60° inclination.

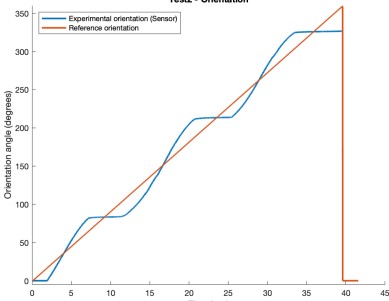

(**f**) Orientation versus time through inertial sensor measurements for 60° inclination.

**Figure 21.** Test 2—Sensor data. Fixed 30°, 45° and 60° inclination for a 360° rotation. The blue line is the experimental data obtained from the encoder and the orange dotted line is the reference.

## 5. Discussion

Simulation and experimental results have been performed to analyze and validate both the design and the proposed model for the cable-driven soft joint.

The simulation results allow the validation of the soft joint through a finite element study. The soft joint was simulated by applying a load of 60 N, which would be the maximum force expected for this prototype. It has made possible to validate the joint structure, ensuring that when maximum loads are applied, the structure does not exceed the elastic limit and does not lose its elasticity.

The experimental tests performed show the behavior of the soft joint system in different situations. Test 1 explores the behavior to reach a target position from a resting position and how the soft link behaves to return to the home position. It is a movement where the

inclination changes with a fixed orientation that does not vary. Test 2 explores the ability to maintain a fixed inclination while gradually varying the orientation.

### 5.1. Results Using the Encoder Sensor

The inclination results, obtained from the encoder during Test 1, show that the experimental inclination reaches the reference inclination, and this is repeated for each of the four requested orientations. We also observed that the higher the requested reference, the longer it takes to reach it.

For the orientation results, the orientation reference is a set of four steps of different sizes. The first is a step of zero amplitude and the experimental orientation is quickly reached. This is because, from the zero-degree inclination position (fully extended joint), reaching any orientation is almost immediate. When the joint is requested to return to the resting position, the experimental orientation remains constant. Meanwhile, the inclination decreases and when it reaches zero, the orientation reaches zero, too. This is why, in this test, the orientation values change so quickly back to zero degrees and the time between the reference orientation and the experimental orientation reaching zero is longer.

### 5.2. Results Using the Inertial Sensor

The data obtained from the inertial sensor show more accurately the real behavior of the end of the soft joint.

The inclination results in Test 1 show that the position of the joint does not reach the reference inclination. The 90° orientation test (downwards, in the sense of gravity) is the one that presents lower errors when tracking the reference. The tests for 0° and 180° orientation angles show higher tracking errors, as shown in the attached video. The kinematics designed for these positions assumes that the length of tendon 1 (lower motor) should not change. These theoretical results, when taken to the experimental field, are not fulfilled because the tendons are not perfectly tensioned, and the two upper wires cause the position rise. This rise is reflected in the orientation that has a negative phase shift when the reference is 0° and a positive phase shift when the reference is 180°.

We also observed that the orientation results do not reach the zero position when the reference is zero. This is because it is difficult to move the orientation to zero due to the fact that the inclination is not exactly zero when returning to the resting position, as the inclination graphs show. This causes a slight inclination while maintaining the same orientation. As discussed above for the encoder data, orientation is very sensitive to inclination.

For the sensor results in Test 2, the inclination graphs show how the experimental inclination does not reach the reference value. However, it should be noted that it has a sinusoidal behavior over time. As in the previous test, the reason for both is that the theoretical behavior of the joint is not the same as the real behavior, because the model assumes aspects such as a continuous curvature, and because there are also other influencing mechanical aspects, such as the precision in the tendon length or the tendon winding in the winches.

This undulatory behavior is observed again in the orientation graphs. However, it can be seen that for angles 90°, 210° and 330° the orientation does not vary, which coincides with the vertices of the soft joint morphology. For these angles, the inclination is maximum. Moreover, when one of the vertices is passed, the opposite tendons cause the variation of orientation, and it takes a little time to change from unwinding to rewinding. This can be seen in the attached videos for this test.

## 6. Conclusions

This work presents a novel approach to soft robotics with the design of a flexible and compact soft joint. It is not only a low-cost prototype, assembled by 3D printing. It also has a morphology that allows better handling of external loads and gravity thanks to its blocking configuration. Actuated by tendons, the proposed design has a morphology with

two main configurations of flexion, which provides more versatility and a flexion limit, unlike previous designs. These characteristics and configurations can be modified through the parameters of the joint morphology, to achieve different fields of work and functionality.

A mathematical model of the inverse kinematics of the soft joint is also presented to obtain the length of the tendons as a function of the morphology and the position (orientation and inclination) of the end of the joint. The modeling of the soft morphology is a complex task, but a simplified and sufficiently accurate kinematic model has been shown. For its validation, the soft link prototype has been built and simulation and experimental studies have been carried out.

According to the capabilities of the solution described and demonstrated throughout the paper, the soft joint proposed in this work shows an improvement over other designs and it could be used for many different applications requiring manipulation of loads. Our main application will be the use of this joint as an arm for the humanoid robot TEO so that the robot can perform manipulation tasks with the use of a gripper connected to the arm tip. There are several uncertainties and mismatches that affect the model of the prototype, especially when this is a low-cost 3D printed solution. For instance, the curvature of the real model is not constant, the tension and length of tendons are not exact, and small variations in the radius of the winches happen after several turns. Despite these facts, the proposed model is accurate enough to represent the kinematics of the system and will allow a later control of the soft joint in closed loop. Further research will lead to reducing these inaccuracies and prototyping effects and to closing the control loop and testing the platform with different loads during manipulation interactions.

## 7. Patents

The technology presented in this paper is under a patent licensing process. A patent entitled "Eslabón para articulación blanda y articulación blanda que comprende dicho eslabón" ("Link for soft articulation and soft articulation comprising such link") and reference number P202030726 (register number 5349) has been presented to the Oficina Española de Patentes y Marcas—OEPM (Spanish Patents Office) (5 July 2020).

**Author Contributions:** Conceptualization, L.N., C.R., C.A.M. and C.B.; methodology, L.N., C.R. and C.A.M.; software, L.N. and C.R.; validation, L.N., C.R. and C.A.M.; formal analysis, L.N. and C.R.; investigation, L.N., C.R., C.A.M. and C.B.; resources, C.A.M. and C.B.; data curation, L.N. and C.R.; writing—original draft preparation, L.N. and C.R.; writing—review and editing, L.N., C.R., C.A.M. and C.B.; visualization, L.N. and C.R.; supervision, C.A.M. and C.B.; project administration, C.A.M.; funding acquisition, C.A.M. and C.B. All authors have read and agreed to the published version of the manuscript.

**Funding:** The research leading to these results has received funding from the project Desarrollo de articulaciones blandas para aplicaciones robóticas, with reference IND2020/IND-1739, funded by the Comunidad Autónoma de Madrid (CAM) (Department of Education and Research), and from RoboCity2030-DIH-CM, Madrid Robotics Digital Innovation Hub (Robótica aplicada a la mejora de la calidad de vida de los ciudadanos, FaseIV; S2018/NMT-4331), funded by "Programas de Actividades I+D en la Comunidad de Madrid" and cofunded by Structural Funds of the EU.

**Institutional Review Board Statement:** Not applicable.

**Informed Consent Statement:** Not applicable.

**Data Availability Statement:** The data presented are available on request from the corresponding author.

**Conflicts of Interest:** The authors declare no conflict of interest.

## Abbreviations

The following abbreviations are used in this manuscript:

| | |
|---|---|
| CAD | Computer-aided Design |
| CDPM | Cable-Driven Parallel Mechanism |
| DLR | Germany's Research Centre for Aeronautics and Space |
| DOF | Degrees of Freedom |
| FDM | Fused Deposition Modeling |
| FEA | Finite Element Analysis |
| FEM | Finite Element Method |
| PC | Personal Computer |
| PLA | Polylactic Acid |
| SMA | Shape Memory Alloy |

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
