# Peer review of "A New Approach of Soft Joint Based on a Cable-Driven Parallel Mechanism for Robotic Applications"

_mathematics, doi:10.3390/math9131468_

Round 1

Reviewer 1 Report

This paper proposes a soft joint with a potential for robotic applications. The geometry and inverse kinematics are studied, simulated, and experimentally tested on a prototype. The paper is overall well structured. This paper should be recommended for publication after the following minor revision points.
(1) Please elaborate on how to get Eqs. (3), (4), and (5). Maybe illustrate with better figures. From Figure 10 (b), it seems that a lot of “triangles” are on a curved surface instead of on a planar surface. If this is the case, I suspect that Eqs. (3), (4), and (5) are at best some approximation, instead of geometrically exact expressions, even under the assumption of a constant bending curvature. If the above suspicion is true, please explicitly state this in the paper.
(2) Since there is a chance that a lot of geometrical relations are simplified, I recommend the authors measure the workspace of the prototype experimentally, and compare it with the theoretical workspace, as in Figure 12, to demonstrate the accuracy of the mathematics.
(3) In Figure 12, the x, y, z coordinates are the coordinates of the center point of the upper surface or what? Please explicitly state.

Reviewer 2 Report

This manuscript presents the design of a soft, tendon-actuated joint with potential applications in soft robotics. The authors perform fine element and inverse kinematic modeling, and some experimental validations to establish the potential use of their design. The manuscript has merit; however, the following comments must be addressed before further consideration:

  1. How did the authors arrive at this particular design, which is shown in Figure 1? Some rationale and detail of the design process must be described.
  2. In line 36, "this article explores this type of design" is not clear. Which of the two designs is being referred? Please be explicit.
  3. The introduction is missing several references. For example:
    1. Line 43-46, cite papers describing morphology driven, tendon-based control, such as https://ieeexplore.ieee.org/document/8481372 and https://doi.org/10.1002/adfm.201808713
    2. line 65-67, examples of servo control of cable-driven soft robotic manipulators, such as https://ieeexplore.ieee.org/document/6696332
  4. In the introduction, the terms "robust" and "robustness" has been used several times, but it is not made clear what is actually being referred to by these words. Do the authors mean strength? Resilience? etc.
  5. Lines 104-114 are unnecessary and should be removed. Same for line 116.
  6. The fact that the soft joint proposed in this work is printed from a flexible polymer should be clearly mentioned in the abstract and also when Figure 1 is referred.
  7. The schematic shown in Figure 2 is not clear. Can this be represented with a more realistic model?
  8. What kind of loading conditions leads to the final configurations shown in Figure 3?
  9. Simulations shown in Figures 7, 8, 9 should be validated experimentally.
  10. For Figure 7, the bending angle due to gravity alone should also be shown clearly, since the figure may be misinterpreted as the joint in perfectly horizontal.
  11. What do the downward red arrows in Figures 7, 8, 9 mean?
  12. What software was used for the finite element simulations? Details of the simulation method and conditions should be provided.
  13. Please provide a more descriptive caption of Figure 14. It may also be helpful to show in each sub-part which length is varied. Moreover, the plots may be easier to interpret if shown as heatmaps instead of 3D surfaces.
  14. Figures 17-28 - please condense these information into lesser number of figures and indicate within the figures what is the experimental setup for each test. Supplementary videos showing the experiments will be very beneficial to the reader.
  15. Some proof of concept application relevant to soft robotics should ideally be shown.

Round 2

Reviewer 2 Report

The authors have satisfactorily addressed all my previous comments. As a result, the manuscript is now suitable for publication in Mathematics.

This manuscript is a resubmission of an earlier submission. The following is a list of the peer review reports and author responses from that submission.